# Adult hospitalizations from immigration detention in Louisiana and Texas, 2015–2018

**Joseph Nwadiuko**[1,2]*, **Chanelle Diaz**[3], **Katherine Yun**[4,5], **Karla Fredricks**[6], **Sarah Polk**[7], **Sural Shah**[8,9], **Nandita Mitra**[10], **Judith A. Long**[1,11]

**1** Division of General Internal Medicine, Perelman School of Medicine, University of Pennsylvania, Philadelphia, Pennsylvania, United States of America, **2** Department of Health Policy and Management, Fielding School of Public Health, University of California, Los Angeles, California, United States of America, **3** Division of General Internal Medicine, Montefiore Health System and Albert Einstein College of Medicine, Bronx, New York, United States of America, **4** Division of General Pediatrics, Children's Hospital of Philadelphia, Pennsylvania, United States of America, **5** University of Pennsylvania Perelman School of Medicine, Philadelphia, Pennsylvania, United States of America, **6** Department of Pediatrics, Section of Global and Immigrant Health, Baylor College of Medicine, Houston, Texas, United States of America, **7** Department of Pediatrics, Johns Hopkins School of Medicine, Baltimore, Maryland, United States of America, **8** Division of Internal Medicine-Pediatrics, University of California, Los Angeles, California, United States of America, **9** Department of Medicine, Olive View-UCLA Medical Center, Los Angeles, California, United States of America, **10** Department of Biostatistics, Epidemiology, and Informatics, University of Pennsylvania, Philadelphia, Pennsylvania, United States of America, **11** Center for Health Equity Research and Promotion, Corporal Michael J. Crescenz VA Medical Center, Philadelphia, Pennsylvania, United States of America

* jnwadiu1@pennmedicine.upenn.edu

**Data Availability Statement:** Data is available by request from the Texas Department of State Health Services (https://www.dshs.texas.gov/thcic/hospitals/inpatientresearchfile.shtm) and the

## Abstract

Poor health conditions within immigration detention facilities have attracted significant concerns from policymakers and activists alike. There is no systematic data on the causes of hospitalizations from immigration detention facilities or their relative morbidity. The objective of this study, therefore, was to analyze the causes of hospitalizations from immigration detention facilities, as well as the percentage of hospitalizations necessitating ICU or intermediate-ICU (i.e, "step-down") admission and the types of surgical and interventional procedures conducted during these hospitalizations. We conducted a cross-sectional study of statewide adult (age 18 and greater) hospitalization data, with hospitalizations attributed to immigration facilities via payor designations (from Immigration and Customs Enforcement) and geospatial data in Texas and Louisiana from 2015–2018. Our analysis identified 5,215 hospitalizations of which 887 met inclusion criteria for analysis. Average age was 36 (standard deviation, 13.7), and 23.6% were female. The most common causes of hospitalization were related to infectious diseases (207, 23.3%) and psychiatric illness (147, 16.6%). 340 (38.3%) hospitalizations required a surgical or interventional procedure. Seventy-two (8.1%) hospitalizations required ICU admission and 175 (19.5%) required intermediate ICU. In this relatively young cohort, hospitalizations from immigration detention were accompanied with significant morbidity. Policymakers should mitigate the medical risks of immigration detention by improving access to medical and psychiatric care in facilities.

Louisiana Department of Health (https://ldh.la.gov/page/2192).

**Funding:** Funding Support: Dr. Nwadiuko was supported by training grant T32HP1002623 from the National Institutes of Health. Further funding was provided from the Matt Slap Pilot Research Award, Conill Endowed Fund, and Office of the Chief of the Division of General Internal Medicine of the University of Pennsylvania. Role of the Funder/Sponsor: One of the authors (JL) provided funding from an unrestricted account. Otherwise, funding sources played no role in the design and conduct of the study; collection, management, analysis, or interpretation of the data; preparation, review, or approval of the manuscript; and decision to submit the manuscript for publication.

**Competing interests:** No authors have any competing interests to report.

## Introduction

Over the last three decades, an exponential increase in detained individuals has overwhelmed the capabilities of immigration jails and prisons to attend to peoples' basic health needs and prevent the spread of infectious diseases. Following the Illegal Immigration Reform and Immigrant Responsibility Act of 1996, the United States government dramatically increased the use of civil detention for non-citizens awaiting determination of their immigration cases. This led to a sharp increase in the number of immigrants in detention from 8,500 in 1996 to over 500,000 people in 2019, one of three of which are within Texas or Louisiana [1–3]. Today, the U.S. immigration detention system, operated by Immigration and Customs Enforcement (ICE) under the Department of Homeland Security (DHS), is the largest national immigration detention system in the world [4]. Customs and Border Patrol (CBP), which operates within 100 miles of the US borders, operates a similarly expansive chain of stations used for holding detainees before they are turned over to ICE.

A DHS Office of Inspector General investigation of several ICE-owned and contracted facilities found "egregious violations" of detention standards [5] including (but not limited to) inadequate medical care, food safety issues, overly liberal segregation practices, and unsanitary conditions [6]. Government whistleblower reports have alleged "grossly negligent" medical care with denials and delays of needed medical and mental health attention [7]. These accounts were confirmed by a US House of Representatives oversight investigation that concluded "immigrants in detention centers operated by for-profit contractors are facing negative health outcomes and even death because of inadequate medical care, poor conditions, understaffing, and delayed emergency care." [8] CBP and ICE's failure to maintain a sanitary environment, adequately immunize detainees, and properly monitor and manage infectious diseases has led to several infectious disease outbreaks [9].

Although mortality amongst immigrant detainees declined precipitously from 2003 and 2015, it has been rising again in recent years [10]. Between April 2018 and September 2020, 38 individuals died in ICE custody; between fiscal year 2019 and 2020 the death rate per 100,000 individuals in ICE detention increased seven-fold, with suicide by hanging as a prominent cause of death [11]. Since 2018, Congress has required CBP and ICE to publicly release all in custody death investigation reports within 90 days of any death. However, beyond case reports from deceased or released detainees, ICE does not provide any data regarding other health outcomes or healthcare quality for its detainees, and in 2016 the US Government Accountability Office found that ICE was not accurately tracking medical utilization by detainees [12].

Detainee mortality case reports do not fully capture morbidity within detained immigrants, and there is little data that fully captures such morbidity within this population. The only known study examining morbidity in immigrant detainees was drawn from a survey of California detention centers in 2013 which demonstrated that 42.5% of all detainees had at least one chronic medical condition [13]. While there are survey studies examining morbidity within non-immigration federal prisons [14], understanding of morbidity within immigration detention facilities is generally limited to case reports shared after detainee death or release [15,16]. Using ICE's unique payor designation and geospatial data information on ICE and CBP facilities, this analysis sought to identify hospitalizations among immigrants detained in federally and privately owned facilities in Texas and Louisiana between 2015 and 2018, including primary diagnoses, primary procedures, and intensive care unit (ICU) and ICU step-down admission rates. This analysis focuses on Texas and Louisiana since a significant portion of detained immigrants are in either state. This period chosen covers both Democratic (Obama) and Republican (Trump) administrations, as well as the period between April and June 2018

during which the Trump administration enforced its "zero-tolerance" border policy, resulting in the separating of 5,500 children from their parents at the US Southern Border.

## Methods

### Ethics statement

The following protocol was approved by the Institutional Review Boards of the University of Pennsylvania, the Texas Department of State Health Services (DSHS), and the Louisiana Department of Health. Formal consent was waived for participants due to concerns with anonymity and feasibility. We also received a Certificate of Confidentiality from the National Institutes of Health.

### Data

See Table 1 for listed data sources. We acquired 2015–2018 hospitalization data from two sources. The first, the Texas Inpatient Research Data File provided by the Texas Healthcare Information Council (THCIC), contains data on discharges from all hospitals with the exception, per THCIC of "hospitals. . .located in a county with a population less than 35,000, or those located in a county with a population more than 35,000 and with fewer than 100 licensed hospital beds and not located in an area that is delineated as an urbanized area by the United States Bureau of the Census. . .exempt hospitals also include hospitals that do not seek insurance payment or government reimbursement." It has been estimated that the Texas Inpatient Data File represents 93–97% of all inpatient discharges in the state [17]. The second, the Louisiana Hospital Inpatient Discharge Database (LHIDD), contains information on all inpatient discharges from licensed hospitals in the state.

Hospitalizations for immigrants in government detention (meaning individuals residing in privately operated or government operated immigration detention facilities) were identified by two mechanisms. First, we looked for ICE's payor designation, which is unique to CBP and ICE detainees. The payor designation was only available for the latter half of 2018 (in Texas) and for one of the five detention facilities in Louisiana between 2015–2018 (LaSalle Detention Center, Jena, Louisiana). As such, we supplemented our search in Texas by employing two sources of geospatial identifiers, both derived from patient-level addresses: nine-digit ZIP (ZIP +4) codes and census bocks. (Neither geospatial identifier was available for use within the Louisiana HIDD.)

ZIP+4 codes are generally specific to one side of a street over a block. This is advantageous for identifying patients residing in often rurally located immigration detention facilities. We

**Table 1. Data sources.**

| Data | Data Source(s) |
| --- | --- |
| Hospitalization/Comorbidity | Texas Inpatient Research Data File<br>Louisiana Hospital Inpatient Discharge Database |
| ZIP +4 Codes | United States Postal Service |
| Census block data | 2010 United States Census |
| Satellite Imagery (to confirm the correct assignment of geospatial identifiers) | Google Earth |
| Detention Facility Information | **Owner, primary care provision, population, length of stay:**<br>US Immigration and Customs Enforcement, via FOIA request from Immigrant Legal Resource Center<br>**National Origin of Detainees:**<br>US Immigration and Customs Enforcement, via Transactional Records Access Clearinghouse of Syracuse University |

examined satellite images of each facility as well as the United States Postal Service (USPS) ZIP Code directory (www.usps.com) to ensure that we did not capture other residential facilities (e.g., private residences) using this method. No immigration detention facilities shared a ZIP +4 with any residences.

ZIP+4 codes were only provided for about one-third of Texas hospitalizations, dependent largely on hospital provision of 9-digit ZIP codes to Texas DSHS. Therefore, this approach was supplemented with patient-level census blocks provided by Texas DSHS. Census blocks are US Census derived micro-geographic units that generally (but not always) cover the area of a city block. Census blocks associated with detention centers occasionally also contain other housing units. Therefore, we located the census block associated with each facility's address, confirming it with visual inspection of satellite imagery. Then, using residential data from the 2010 US Census, we obtained the number of surrounding housing units in the same census block. (2020 US Census housing unit data had not been released at the time of writing.) While full addresses were not provided in the Texas Inpatient Data File, the state provided data on the accuracy of patient listed addresses received from hospitals; only hospitalizations with linked addresses accurate to the center of a building were included.

We then separated all hospitalizations into those for which we had "excellent confidence" that the patient in question originated in a CBP station or ICE-related facility (i.e., linked with ICE payor information, a ZIP4+ code matching a facility, or a census block linked to a facility with no other housing units), "high confidence" (i.e., facilities linked to a census block with 1–10 housing units), "good confidence" (i.e., facilities linked to census blocks with 11–25 housing units), "poor confidence" (i.e., facilities linked to census blocks with 26–100 housing units) and "very poor confidence" (i.e., facilities linked to census blocks with greater than 100 housing units).

To maximize accuracy without excessively limiting the number of facilities covered, our primary analyses focus on Louisiana hospitalizations (all of which were pre-selected using ICE's payor designations) and Texas hospitalizations meeting the following criteria: 1) ICE payor designation; or 2) "excellent confidence" or "high confidence" that the patient in question originated in a facility that was fully occupied by immigrant detainees based upon the patient-level geospatial identifier. ICE Facilities identified as fully occupied by immigrant detainees included Service Processing Centers, Contract Detention Facilities, Family Detention Centers, or facilities under Dedicated Intergovernmental Service Agreements. Other facilities are contracted to ICE from city and county governments, and these facilities often house city and county prisoners who are not being held for immigration-related offenses, in addition to immigration detainees.

Hospitalizations were excluded if patient age was less than 18 or if payor data showed a payor other than ICE, Self-Pay, or Charity Care. Self-pay and Charity Care payor designations were included because ICE has released patients from detention just before or while they are hospitalized with critical illness, potentially shifting financial responsibility for the hospitalization to the patient or the safety net [18]. Facility level information (e.g. average detainee length of stay, population, medical staffing) was also obtained from records provided by ICE after a Freedom of Information Act (FOIA) request by the Immigrant Legal Resource Center [19]; this data is also used to compute hospitalization rates during the 2016 and 2017 fiscal years, as described in Supplementary Analysis. The nationalities of detained immigrants were obtained from the Detention Dataset of the Transactional Records Access Clearinghouse of Syracuse University [20]. Equivalent facility-level data were not available for individual CBP facilities. However, it is known that all CBP facilities are federally owned and operated, and the average length of stay for adults at the US Southwestern Border in fiscal year 2015 was 39.2 hours

(standard deviation 128.3), with variation by time of year, apprehending station, and detainee nationality [21].

## Outcomes and analysis

We describe three outcomes: 1) principal diagnoses of admissions; 2) principal invasive procedures during admissions; 3) and the percentage of hospitalizations that involved stays in intensive care units (ICUs) or intermediate-intensive care units. We also determined if suicidality or self-harm was mentioned as a primary or secondary diagnosis given the relatively high amount of attributable mortality to suicide within detention facilities [22].

Census blocks were assigned using ArcMap 10.7.1 (ESRI, Redlands, California), satellite images were viewed using Google Earth (Mountain View, CA), and statistical analysis was done in STATA 17 (College Station, Texas).

## Supplementary analysis

We carried out three supplementary analyses. The first analysis describes hospitalization data only from hospitalizations that we had "excellent confidence" originated from immigration detention facilities. Secondly, we described hospitalization data only from hospitalizations that we had "good confidence" originated from immigration detention facilities, which was not included in the principal analysis. Finally, we estimate hospitalization rates per 1000 prisoner years from ICE detention centers from fiscal years 2016 and 2017, reflecting the population data available from the FOIA records provided by ICE to the Immigrant Legal Resource Center [19]. (We generated facility-level hospitalization rates from counts of facility-level hospitalizations, the total number of book-ins (i.e., entrants) per facility for each fiscal year (as the base population), and the facility-level average length of stay. A general rate was created as the average of facility level rates, weighted by facility-level book-in counts. This does not include CBP hospitalizations, due to lack of site-specific population data).

## Statistical analysis

We analyzed our results descriptively, with a particular focus on primary diagnoses related with hospitalizations, procedures, and admissions to advanced care (ICU or intermediate-ICUs).

## Results

We accessed data extracts for 12,356,447 hospitalizations from Texas and 68 pre-selected hospitalizations from Louisiana (based on ICE's payor designation) (see S1 Fig for diagram). Of the Texas hospitalizations, 5,215 were identified by census blocks (n = 4,350), ICE's payor designation (n = 107), or ZIP+4 codes (n = 758). Of the 4,350 hospitalizations identified by census blocks, 27 were excluded due to lack of specificity to the level of a building and 3,311 were excluded due to originating from less than "high confidence" census blocks, leaving 1,012 census block identified hospitalizations. Out the 1,770 hospitalizations identified by census blocks (after exclusion) or ZIP+4 codes, 753 were excluded due to being from facilities that were not fully dedicated to holding immigration detainees, and 174 were excluded due to having a payor other than ICE, Self-Pay, or charity care, leaving 1,018 hospitalizations. Ninety-five were excluded due to age less than 18 and 36 were excluded as hospital transfers.

Our primary analysis identified 887 hospitalizations from 17 CBP and 9 ICE facilities. The ICE facilities captured in this analysis represented 97.7% of all book-ins into ICE facilities fully occupied by immigrant detainees between 2015 and 2017 [19]. Of the 887 hospitalizations, 100

came from CBP facilities, 724 from ICE facilities, and 63 identified by the ICE payor code for which origin site is unknown. Six hundred and forty-one patients (72.7%) were between the ages of 18 and 45 and 210 were female (23.6%). The hospitalization rate from ICE facilities in fiscal years 2016 and 2017 were 39.6 per 1,000 prisoner-years and 39.3 per 1,000 prisoner-years, respectively. The median length of inpatient admission for all patients was 5 days (inter-quartile range 2, 9). 363 patients (40%) had a comorbid diagnosis of asthma, hypertension, diabetes, COPD, chronic bronchitis, or depression upon admission. Of patients who were admitted from ICE facilities, most came from facilities that were publicly owned (number from privately owned facilities, 250, 30.5%) but all facilities were operated by a government contractor. All patients in ICE custody came from facilities with an active medical provider on site (Table 2), with primary care largely delivered by the federal government (687, 94.9%). Twenty-two percent of persons detained in ICE facilities were Mexican, 18% were from Honduras,16% were from El Salvador, and 15% were from Guatemala. Among the nine ICE detention facilities represented in this sample, the average length of stay per detainees was 23 days (standard deviation 14). (Table 2) The average length of stay in all Customs and Border Patrol facilities at the US Southwestern Border in the fiscal year 2015 was 39.2 hours (standard deviation 128.3) according to reports [21].

**Table 2. Description of patients and originating facilities.**

| N | 887 |
|---|---|
| **Patient Age (mean, standard deviation)** | 36.8 (13.7) |
| **Patient Age (%)** | |
| 18–25 | 209 (23.6) |
| 25–34 | 228 (25.7) |
| 35–44 | 204 (23.0) |
| 45–54 | 132 (14.8) |
| 55–64 | 88 (9.9) |
| 65+ | 26 (2.9) |
| **Sex (%)** | |
| Female | 210 (23.6) |
| Male | 677 (76.3) |
| **Race (%)** | |
| Hispanic | 609 (68.7) |
| Non-Hispanic White | 107 (12.1) |
| Non-Hispanic Black | 39 (4.4) |
| Non-Hispanic Other | 116 (13.1) |
| **Year** | |
| 2015 | 177 (20.0%) |
| 2016 | 205 (23.1%) |
| 2017 | 204 (23.0%) |
| 2018 | 301 (33.9%) |
| **State** | |
| LA | 68 (7.7%) |
| TX | 819 (92.3%) |
| **Co-morbidities** * | |
| Asthma | 21 (2.4%) |
| Chronic Kidney Disease | 89 (10.0%) |
| COPD/Chronic Bronchitis | 16 (1.8%) |
| Diabetes | 118 (13.3%) |

(*Continued*)

**Table 2.** (Continued)

| | |
|---|---|
| Depression | 63 (7.1%) |
| Hypertension | 169 (19.1%) |
| **Type of Facility** | |
| Customs and Border Patrol | 100 (12.1%) |
| Immigration and Customs Enforcement | 724 (81.6%) |
| Unknown | 63 (7.1%) |
| **Medical Provider Onsite** [†] | 760 (82.3%) |
| **ICE Facility Health Care Operator** [†,‡] | |
| Private Contractor | 37 (5.1%) |
| ICE Health Service Corps | 722 (94.9%) |
| **ICE Facility Operator**[†] | |
| Ahtna, Incorporated | 318 (43.9%) |
| CoreCivic | 33 (4.5%) |
| The GEO Group | 296 (40.9%) |
| GPS-ASSET | 50 (6.9%) |
| LaSalle Management Company, LLC | 27 (3.7%) |
| **ICE Facility Owner** [†] | |
| City/County | 136 (18.7%) |
| CoreCivic | 3 (0.4%) |
| The GEO Group | 218 (30.1%) |
| Immigration and Customs Enforcement | 368 (50.8%) |
| **ICE Facility Level Data, Fiscal Year 2015–2017** [†] | |
| N | 9 |
| Average Daily Population, mean (SD) | 854 (409.8) |
| Average Annual New Detainees Booked, mean (SD) | 17628 (13596.1) |
| Average Length of Stay (days), mean (SD) | 23 (14.2) |

ICE = Immigration and Customs Enforcement.

*Provided by primary and secondary diagnosis data.

† Data only available for ICE Detention Centers, not Customs and Border Patrol Facilities. Data on ICE Detention Facilities was provided by Immigration and Customs Enforcement after a FOIA request by the Immigrant Legal Resource Center.

‡ Categories are collapsed for cell sizes greater than 15 for patient privacy in accordance with the Data Use Agreement with Texas Department of State Health Services.

The most common inpatient principal diagnoses were related to infectious causes (207, 23.3%), driven primarily by skin infections and abscesses, active tuberculosis, and pneumonia (Table 3). Psychiatric diseases made up the second most common cause (147, 16.6%), with hospitalizations largely related to mood disorders (e.g., anxiety, depression, and post-traumatic stress syndrome), followed by bipolar disease. Fourty-eight (5.4%) admissions had as a primary or secondary diagnosis suicidal ideation or intentional self-harm. Seventy-six percent of psychiatric hospitalizations and 72% of self-harm related hospitalizations originated from a single facility, the South Texas Detention Complex. Other common causes of morbidity were related to cardiovascular disease (95, 10.7%) and gastrointestinal disease (81, 9.1%). Hospitalizations from ICE custody were largely related to infectious disease (179, 24.7%) and psychiatric disease (145, 20.3%). Meanwhile, hospitalizations from CBP custody were predominantly related to trauma and toxic exposure (17, 17.0%); heat exposure, syncope, and rhabdomyolysis (16, 16.0%); infectious diseases (16, 16.0%); and obstetric presentations (16, 16.0%). About one in

**Table 3. Principal diagnoses associated with hospitalizations[a].**

| Categories | Count | Percent |
|---|---|---|
| **Cardiovascular Disease** | **95** | **10.7** |
| *Specific Causes* | | |
| Chest pain without demonstrated ischemia | 20 | 21.1 |
| Hypertension and Hypertensive Complications | 27 | 28.4 |
| Arrhythmias and Congestive Heart Failure | 15 | 15.8 |
| Unstable Angina and Myocardial Infarction | 29 | 30.5 |
| Other | 4 | 4.2 |
| **Complications of interventions and surgeries** | **16** | **1.8** |
| **Endocrine Disease** | **21** | **2.4** |
| *Specific Causes* | | |
| Diabetes Mellitus | 18 | 85.7 |
| Other | 3 | 14.3 |
| **Gastrointestinal Disease** | **81** | **9.1** |
| *Specific Causes* | | |
| Appendicitis | 22 | 26.8 |
| Hepatobiliary Disease | 28 | 34.1 |
| Other | 31 | 38.2 |
| **Gynecologic and Genitourinary Disease** | **17** | **1.9** |
| **Heat exposure, Rhabdomyolysis, and Syncope** | **437** | **5.3** |
| **Hematologic and Oncologic Disease** | **17** | **1.9** |
| **Infectious Disease** | **207** | **23.3** |
| *Specific Causes* | | |
| Active Tuberculosis | 25 | 12.1 |
| GI and GU infections | 16 | 7.7 |
| Nontuberculosis Pneumonias | 31 | 15.0 |
| Skin Infection/Abscess | 53 | 25.6 |
| Sepsis from other causes | 39 | 18.9 |
| Other | 43 | 20.8 |
| **Neurological Disease** | **37** | **4.2** |
| *Specific Causes* | | |
| Epilepsy | 19 | 51.4 |
| Other | 18 | 48.6 |
| **Obstetric Presentations** | **29** | **3.3** |
| **Psychiatric Disease** | **147** | **16.6** |
| *Specific Causes* | | |
| Anxiety, Depression, and other Mood Disorders | 58 | 39.5 |
| Bipolar Disease | 32 | 21.8 |
| Psychosis and Delusion not otherwise specified | 16 | 10.2 |
| Schizophrenia spectrum | 26 | 17.7 |
| Other | 15 | 10.2 |
| **Pulmonary Disease** | **35** | **4.0** |
| *Specific Causes* | | |
| Solitary Pulmonary Nodules & other Abnormal Findings | 16 | 45.7 |
| Other | 19 | 54.3 |
| **Renal disease** | **57** | **6.4** |
| *Specific Causes* | | |
| Acute Kidney Injury | 29 | 50.9 |

(*Continued*)

**Table 3.** (Continued)

| Categories | Count | Percent |
|---|---|---|
| Hypo- or Hypervolemia and Electrolyte Derangements | 28 | 49.1 |
| **Rheumatologic and Musculoskeletal Disease** | **15** | **1.7** |
| **Trauma and Toxic Exposure** | **49** | **5.5** |
| **Other** | **17** | **1.9** |
| **Total** | **887** | **100.0** |

*Categories are collapsed for cell sizes greater than 15 for patient privacy in accordance with the Data Use Agreement with Texas Department of State Health Services.

three admissions involved an invasive procedure (340, 38.3%), most commonly incision and drainage (54, 6.0%) (most prominently for skin, gastrointestinal, and pulmonary tissue) and orthopedic surgical procedures (40, 4.5% of hospitalizations) (Table 4).

About one in every 4 admissions (245, 27.6%) required an ICU (72, 8.1%) or intermediate-ICU (175, 19.5%) admission; the average age of admitted patients was 40 (standard deviation 14.4). Cardiovascular causes made up the most common cause of higher-level admissions overall (71, 29.0%); the average of admitted patients in this group was 50 (standard deviation 11.1). 38 (38%) patients admitted from CBP custody required intermediate ICU or ICU care, compared to 170 (23.5%) of those admitted from ICE custody. Over the period covered by this study, rates of admission rose for ICUs (8.5% (15) in 2015 to 11.3% (34) in 2018) and intermediate ICUs (15.8% (28) in 2015 to 22.9% (70) in 2018) (Fig 1). Analysis of hospitalizations meeting "excellent confidence" criteria are notable for revealing a higher ICU/intermediate-ICU admission rate of 33.2% (S1 and S2 Tables). Analysis of hospitalizations meeting "good confidence" criteria show a similar set of results as those described (S3 and S4 Tables).

**Table 4. Surgeries and procedures*.**

| Procedures | Count | Percent |
|---|---|---|
| Appendectomy | 21 | 2.28 |
| Cholecystectomy | 22 | 2.38 |
| Dialysis and Renal Replacement Therapies | 15 | 1.63 |
| Endoscopic and Interventional Gastrointestinal Procedures | 15 | 1.63 |
| Incision and Drainage | 54 | 5.85 |
| Orthopedic surgical procedures | 40 | 4.33 |
| Peripheral and Cardiac Bypass Surgery and Catherization, Cardioversion, and Insertion of Cardiac Devices | 33 | 3.58 |
| Thoracentesis, Pulmonary Biopsy, and Bronchoscopy | 17 | 1.84 |
| Transfusion of Blood Products | 15 | 1.63 |
| Other Abdominal Surgeries | 18 | 1.95 |
| Other dermatologic and subcutaneous surgeries | 21 | 2.28 |
| Other procedures and surgeries | 71 | 7.69 |
| None | 581 | 62.95 |
| Total | 887 | 100 |

*Categories are collapsed for cell sizes greater than 15 for patient privacy in accordance with the Data Use Agreement with Texas Department of State Health Services.

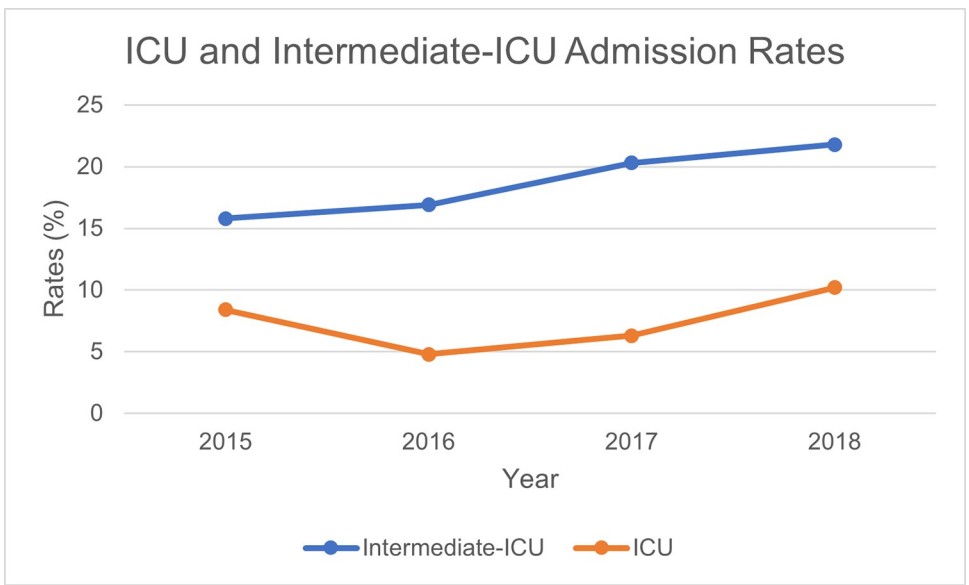

**Fig 1. ICU and intermediate-ICU admission rates.**

## Discussion

The following analysis finds that among hospitalized immigrant detainees in Texas and Louisiana there is a disproportionate burden of infectious and psychiatric illness and a high need for advanced care. This study represents one of the first attempts at understanding the morbidity burden among immigrant detainees.

Our study shows that several indicators of illness severity may be higher than expected. First, ICU admission rates in this group approach the national average of 26.9%, even though this group was younger than most hospitalized populations (72% of hospitalizations in our sample were from patients between 18 and 44, compared to 28.8% of adult hospitalizations nationally) [23,24]. In addition, morbidity is generally lower among new immigrants than US born individuals, and thus we might have expected even lower ICU admission rates [25]. Second, the preponderance of some procedures, such as incision and drainage, suggest that illnesses (such as infections) have often progressed to severe conditions (e.g., abscesses) requiring procedural intervention by the time patients are hospitalized. This is accompanied by a very low hospitalization rate among ICE detainees: the hospitalization rate of 39 per 1,000 prisoner-years (for the years in which we had census data on ICE detainees) is lower than the average of 54 per 1,000 prisoner-years for non-immigrant incarcerated persons [26] and 67 per 1,000 for undocumented immigrants [27]. More data are needed to better understand the higher-than-expected ICU admission rate, lower-than-expected hospitalization rates, and large number of certain procedures, and it is possible that immigrants are arriving to immigration detention facilities at advanced stages of illness after travel across the Southern US Border or due to lack of healthcare access outside of detention facilities. Another possibility however is that immigration detention facilities may not be properly managing conditions prior to hospitalization or, potentially, they are delaying care until the patient worsens enough to warrant hospitalization, as has been suggested by multiple governmental reports [6–9].

There are several possible mechanisms behind this phenomenon. While ICE has created a set of standards for healthcare delivery, those standards have not been rigorously enforced, with one study finding a violation of ICE's medical standards in 78.2% of all deaths in ICE

custody, largely involving delays and denials in receiving appropriate care [15,28]. According to leaked DHS internal reports, many instances of avoidable deaths or morbidity have been reported to ICE's medical senior management without any corrective action [29]. CBP, on the other hand, had no internal medical protocols until 2018 and had a paucity of medical personnel on site, with one official stating in congressional testimony that up to three stations shared a single emergency medical technician [30].

The high burden of psychiatric illness found in this analysis, as well as suicidal ideation and self-harm, cannot be overlooked. Asylum seekers and refugees from Central America often have experienced trauma before or during their migration journeys to the United States [31]. Furthermore, studies have estimated the prevalence of post-traumatic stress disorder among refugees to be high as 36%, and psychosis rates among immigrants may be higher than native populations, driven by social disadvantage pre- and post-migration [32,33] There has also been much written about the psychiatric impacts of immigration detention and incarceration generally, with studies demonstrating detention as an independent risk factor for anxiety, depression, and post-traumatic stress disorder [34]. Furthermore, leaked internal reports and external analyses have conveyed how immigration detention centers have been poorly trained to handle psychiatric emergencies, often remanding patients to solitary confinement instead of escalating their medication regimen [35]. Finally, while ICE does have internal standards to prevent suicide attempts in particular, implementation of suicide awareness training at detention sites has been poor, which may account for the high level of detainee suicide attempts [22].

ICE's use of solitary confinement merits further attention. A report by the Project on Government Oversight noted that out of 6,559 instances of solitary confinement in ICE facilities between 2016 and 2018, 39% were for individuals with a history of mental illness; among those with mental illness, 38% were in solitary confinement for 15 days or more [35]. In our dataset, 7% of psychiatric hospitalizations and 72% of self-harm related hospitalizations came from the South Texas Detention Complex. Of note the South Texas Detention Complex had the highest number of solitary confinement episodes among ICE facilities in Texas; 80% of those placed in confinement had a history of mental illness [35]. While it is unclear whether solitary confinement practices were temporally related to hospitalizations, this spatial correlation might suggest that patients' needs and healthcare processes might be misaligned.

There are divergences between the burden of disease between hospitalizations from immigrant and non-immigrant detention. A study from Texas showed that among prison hospital discharges across the state, 22.4% were related to cardiac related illness and 18% were related to gastrointestinal illness, with the other admissions related to other medical causes [36]. This does not include psychiatric disease since state prisons often triage psychiatric illness to internal psychiatric units. However, emergency department (ED) admission data from Tennessee and New York State show that 4% and 13% of presenting diagnoses were psychiatrically related [37,38] This is likely, however, due to differences in baseline co-morbidities between immigrant and local populations: Among non-immigrant patients who presented to ED from prisons, 35% have a comorbidity of hypertension, 16% have a comorbidity of coronary artery disease, 11% have a comorbidity of diabetes, and 45% have a comorbidity of psychiatric illness [38].

There are several policy implications for this analysis. First, better adherence to ICE's own detention standards might help reduce avoidable morbidity and mortality in detention centers. In particular, patients deserve rapid and thorough triage at illness presentation within facilities. Second, given the frequency of admissions for psychiatric illness, DHS should robustly assess the mental health risk factors of immigrants and take them into consideration when considering whether to place individuals into detention. Inasmuch as other causes of hospitalization

are related to exacerbations of chronic disease, thorough surveillance of health risk factors during detention is also critical. Finally, the federal government should increase the availability of data relating to immigration detention hospitalizations, which have a common payer via ICE. To our knowledge, immigrant detainees are the only cohort of patients with federally sponsored healthcare for which there is no federally released health data. Allowing more systematic access to health data—paired with detention characteristics data (e.g., length of detention stay) —will help generate evidence-based interventions to prevent morbidity and mortality in facilities.

## Limitations

There are several limitations to this analysis. First, this analysis is cross-sectional, and no causality can be claimed. Second, we are unable to assess the complete rate of hospitalizations in this cohort since complete detention population data is not available through the entirety of the study period, although partial hospitalization rates are listed. Third, the use of "high confidence" census blocks might include data from non-detainees, although we believe that, if present, it would bias our data towards decreased severity: examination of "excellent confidence" hospitalizations showed a higher advanced care admission rate than the combined sample.

Fourth, this study does not reflect modern comorbidities and structural changes related to the severe acute respiratory syndrome coronavirus-2. Since the beginning of the pandemic, ICE detention centers saw historic depopulation with daily population falling from 38,537 in February 2020 to a nadir of 13,258 in February 2021. However, since Spring 2021, counts have begun to rise again, and as of December 2021 there were 20,623 detainees in ICE custody [20]. Furthermore, even if vaccination is proven to be effective at reducing the spread of virus in incarceration settings, in the absence of further systemic changes, it is possible that the outcomes described in this paper may persist in a COVID- and post-COVID era. Beyond vaccination, immigration detention should be structurally reformed to sustainably decrease morbidity generation.

## Conclusion

Hospitalizations from immigration detention are associated with significant morbidity. Many hospitalizations are related to infectious and psychiatric disease and involve ICU stays as well as an invasive procedure. More research is needed to determine the etiology of these findings, especially related to the quality (or lack thereof) of medical and mental health services provided in immigration detention facilities. Ongoing advocacy is needed to ensure that detained immigrants receive fair and humane treatment, including access to appropriate and timely health care.

## Supporting information

**S1 Fig. Identification of immigration detention-related hospitalizations.**
(DOCX)

**S1 Table. Principal diagnoses associated with hospitalizations with "excellent confidence" of coming from a detention facility fully occupied by immigrants.**
(DOCX)

**S2 Table. ICU and intermediate-ICU admissions associated with hospitalizations with "excellent confidence" of coming from a detention facility fully occupied by immigrants.**
(DOCX)

**S3 Table. Principal diagnoses associated with hospitalizations with "good confidence" of coming from a detention facility fully occupied by immigrants.**
(DOCX)

**S4 Table. ICU and intermediate-ICU admissions associated with hospitalizations with "good confidence" of coming from a detention facility fully occupied by immigrants.**
(DOCX)

## Acknowledgments

The authors wish to acknowledge comments from Dr. Arturo Bustamante-Vargas, data support from the Texas Health Care Information Collection and the Louisiana Department of Health, as well as data infrastructure support from the Leonard Davis Institute's Health Services Research Data Center.

## Author Contributions

**Conceptualization:** Joseph Nwadiuko, Katherine Yun, Karla Fredricks, Judith A. Long.

**Data curation:** Joseph Nwadiuko.

**Formal analysis:** Joseph Nwadiuko.

**Funding acquisition:** Joseph Nwadiuko.

**Investigation:** Joseph Nwadiuko, Chanelle Diaz, Katherine Yun, Karla Fredricks, Sarah Polk, Sural Shah, Nandita Mitra, Judith A. Long.

**Methodology:** Joseph Nwadiuko, Chanelle Diaz, Katherine Yun, Karla Fredricks, Sarah Polk, Sural Shah, Judith A. Long.

**Project administration:** Joseph Nwadiuko.

**Resources:** Judith A. Long.

**Supervision:** Katherine Yun, Judith A. Long.

**Validation:** Joseph Nwadiuko.

**Visualization:** Joseph Nwadiuko.

**Writing – original draft:** Joseph Nwadiuko, Chanelle Diaz.

**Writing – review & editing:** Joseph Nwadiuko, Katherine Yun, Karla Fredricks, Sarah Polk, Sural Shah, Nandita Mitra, Judith A. Long.

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
