## [Decision Letter · Decision Letter 0]

28 Feb 2022

PGPH-D-22-00070

Adult Hospitalizations from Immigration Detention in Louisiana and Texas, 2015-2018

Dear Dr. Nwadiuko,

Thank you for submitting your manuscript to PLOS Global Public Health. After careful consideration, we feel that it has merit but does not fully meet PLOS Global Public Health’s publication criteria as it currently stands. Therefore, we invite you to submit a revised version of the manuscript that addresses the points raised during the review process.

EDITOR: The reviewers have raised many important points and suggestions. We're interested in evaluating a revised version that addresses them listed below.

We look forward to receiving your revised manuscript.

Kind regards,

Young-Rock Hong

Academic Editor

Journal Requirements:

1. We see that your study includes live participants, but you have not included an Ethics Statement. Please update your manuscript file to include an Ethics Statement subsection to your Materials and Methods section. It should include:

i) The full name(s) of the Institutional Review Board(s) or Ethics Committee(s)

ii) The approval number(s), or a statement that approval was granted by the named board(s) 

iii) (for human participants or donors) - A statement that formal consent was obtained (must state whether verbal/written) OR the reason consent was not obtained (e.g. anonymity)

2. In the online submission form, you indicated that "Data is available from the Texas Department of State Health Services and the Louisiana Department of Health.". All PLOS journals now require all data underlying the findings described in their manuscript to be freely available to other researchers, either 1. In a public repository, 2. Within the manuscript itself, or 3. Uploaded as supplementary information.

Additional Editor Comments (if provided):

Reviewers' comments:

Reviewer's Responses to Questions

**Comments to the Author**

1. Does this manuscript meet PLOS Global Public Health’s publication criteria? Is the manuscript technically sound, and do the data support the conclusions? The manuscript must describe methodologically and ethically rigorous research with conclusions that are appropriately drawn based on the data presented.

Reviewer #1: Partly

Reviewer #2: Yes

Reviewer #3: Yes

Reviewer #4: Yes

2. Has the statistical analysis been performed appropriately and rigorously?

Reviewer #1: Yes

Reviewer #2: Yes

Reviewer #3: Yes

Reviewer #4: Yes

3. Have the authors made all data underlying the findings in their manuscript fully available (please refer to the Data Availability Statement at the start of the manuscript PDF file)?

Reviewer #1: Yes

Reviewer #2: Yes

Reviewer #3: Yes

Reviewer #4: No

4. Is the manuscript presented in an intelligible fashion and written in standard English?

Reviewer #1: Yes

Reviewer #2: Yes

Reviewer #3: Yes

Reviewer #4: Yes

5. Review Comments to the Author

Reviewer #1: This study touches upon one of the most important public health issues, increasing rates of immigrant detention and the consequent health issues in the US. The study very strongly highlights the need for more data in this important area of research.

The authors have described in detail the methodology; however, a couple of concerns need to be addressed.

1. The authors included patients classified as “high confidence” in their primary analysis. This means patients from facilities linked to a census block with 1-10 housing units were included in the analysis. This leads to a greater degree of loss of confidentiality as fewer number of housing units may reveal the patient identity much easily. In this context, it will be interesting to know who (authors or non-authors) were involved in using the satellite images to identify the centers versus houses, and the mechanism used to avoid identifying the houses to protect confidentiality.

2. How much change will happen in the overall results, if only “excellent confidence” cases are analyzed? Is it possible to restrict analysis only to this group?

Other comments:

1. While the authors’ observation that the severity of illness may be higher than expected may be valid, we need more objective data to substantiate the claim. We do not know the duration stay in the center before these patients were referred to health facilities or diagnosed with a condition or underwent surgery. It may also be that they had underlying conditions which were detected once they reached the center. Therefore statements like “there is a tradeoff between a low willingness for facilities to hospitalize patients and a high severity of illness upon hospital admission” may be avoided/modified.

2. The very high number of appendicitis and appendicectomy in the group needs more explanation.

3. A further breakup of the 18-45 age groups into smaller groups (at least splitting the very young age groups out) may be useful. This may help to further clarify the high rate of utilization of cardiac care.

Reviewer #2: Clear and important research questions that is properly answered in the study. This is a novel topic that has not been researched previously, and of great importance and implications. There are no fundamental flaws in the design, and has strong technical rigour.

The papers is well-written, clear and easy to read. Tables and figures are clear and simplified. Abstract is strong, clear and summarises the paper clearly. Methods and results are accessible, justified and clearly presented, they support the conclusion.

Minor comments:

- Include more detail in methodology on statistical analysis used and justification.

- Conclusion requires more details on implications of the study results, how and what can be done with the results, e.g. for policymakers, and/or suggestions for future research to compliment this study.

Reviewer #3: A very nice and well-written quantitative piece on hospitalizations stemming from detention centers in LA and TX. It is an important global health piece that will be an excellent compliment to qualitative explorations and government reports of similar topic.

I have two substantive suggestions:

1. Given the available data from ICE and CBP can the authors provide some information about the detainees in the centers the hospital data is coming from? I assume that they are primarily from Mexico/Northern Triangle countries, but do we have more information (country of origin, language spoken, etc.)?

2. The authors should engage more with implications and connections to larger picture of migration in their discussion. What can this data tell us about the conditions from which migrants are leaving and how that impacts morbidities and mortalities in detention. What does ICE and CBP need to be better prepared to deal with? For example, in the paragraph on the burden of psychiatric illnesses (lines 251-259), a sentence or two on the conditions (violence, poverty, etc) that prompt much of the migration would parallel the data and strengthen the argument that ICE & CBP are not doing enough to meet these acute medical needs.

Minor suggestions:

Make the relevance of LA and TX earlier in the piece. At first glance it seems like an unlikely selection of states. I would move the statement that 1 in 3 detainees are in TX and LA from parentheses and move it up higher in the introduction.

Footnote 1: the footnote was published in 1999 but is used to reference data from 2019. I tried to visit the link (out of curiosity of the stats) and URL didn't work.

Reviewer #4: This study tabulates hospitalization data of immigration detention centers in Texas (the vast majority of the data) and Louisiana during the years 2015-2018, focusing on fully immigrant-occupied facilities. The authors do not make their curated data set available. I appreciate the effort required to rigorously identify facilities meeting their criteria for being fully immigrant-occupied, while excluding locally sourced hospitalizations. The authors describe in great detail what is more clearly understood in the tables. [As a minor point, the heading the second column in Tables 2 and 3 should be Count.]

While I understand the significance of reporting these data from immigrant-occupied facilities, little can be discerned---beyond conjecture---in the absence of comparative data. I suggest that the authors add hospitalization data from regular state penal institutions; given the minor contribution of the Louisiana data, focus on Texas. From the standpoint of public health, comparisons of the fully immigrant-occupied and regular state penal facilities would better identify clear areas of concern. The authors commented on tuberculosis, psychiatric disease, and skin abscesses as being of particular concern, but in the absence of comparative data it is challenging to call attention to poor facility management. I'm left with the question: Where do these results fall among other studies of hospitalization of prisoner populations.

6. PLOS authors have the option to publish the peer review history of their article (what does this mean?). If published, this will include your full peer review and any attached files.

**Do you want your identity to be public for this peer review?** For information about this choice, including consent withdrawal, please see our Privacy Policy.

Reviewer #1: **Yes: **Shaffi Fazaludeen Koya

Reviewer #2: No

Reviewer #3: No

Reviewer #4: No

---

## [Decision Letter · Decision Letter 1]

24 Jun 2022

Adult Hospitalizations from Immigration Detention in Louisiana and Texas, 2015-2018

PGPH-D-22-00070R1

Dear Dr Nwadiuko,

We are pleased to inform you that your manuscript 'Adult Hospitalizations from Immigration Detention in Louisiana and Texas, 2015-2018' has been provisionally accepted for publication in PLOS Global Public Health.

Best regards,

Tharani Loganathan, MD, MPH, DrPH

Academic Editor

Reviewer Comments (if any, and for reference):

Reviewer's Responses to Questions

**Comments to the Author**

1. If the authors have adequately addressed your comments raised in a previous round of review and you feel that this manuscript is now acceptable for publication, you may indicate that here to bypass the “Comments to the Author” section, enter your conflict of interest statement in the “Confidential to Editor” section, and submit your "Accept" recommendation.

Reviewer #1: All comments have been addressed

Reviewer #2: All comments have been addressed

2. Does this manuscript meet PLOS Global Public Health’s publication criteria? Is the manuscript technically sound, and do the data support the conclusions? The manuscript must describe methodologically and ethically rigorous research with conclusions that are appropriately drawn based on the data presented.

Reviewer #1: Yes

Reviewer #2: Yes

3. Has the statistical analysis been performed appropriately and rigorously?

Reviewer #1: Yes

Reviewer #2: Yes

4. Have the authors made all data underlying the findings in their manuscript fully available (please refer to the Data Availability Statement at the start of the manuscript PDF file)?

Reviewer #1: Yes

Reviewer #2: Yes

5. Is the manuscript presented in an intelligible fashion and written in standard English?

Reviewer #1: Yes

Reviewer #2: Yes

6. Review Comments to the Author

Reviewer #1: Thank you for addressing previous review comments. This has improved the manuscript. This is an important topic to be studied in detail, however, this first step descriptive paper needs to be published.

Reviewer #2: n/a

7. PLOS authors have the option to publish the peer review history of their article (what does this mean?). If published, this will include your full peer review and any attached files.

**Do you want your identity to be public for this peer review?** For information about this choice, including consent withdrawal, please see our Privacy Policy.

Reviewer #1: **Yes: **Shaffi Fazaludeen Koya

Reviewer #2: No
